# SILA: Enhancing Long-Context Retrieval Capability of Linear Attention via Selective Ignoring

## Abstract

Linear attention models have recently emerged as computationally efficient alternatives to Transformers. Despite competitive performance on general commonsense tasks, they still struggle to match Transformers on long-context retrieval tasks. In this work, we re-examine linear attention models from the perspective of memory writing. We propose that enabling linear attention models to learn **selective ignoring** provides a promising approach to addressing long-context retrieval tasks under fixed memory capacity. Guided by this principle, we demonstrate how to interpret and intervene in the behavior of linear attention models, thereby revealing the true retrieval capabilities of popular models. Informed by these observations, we introduce Selective Ignoring Linear Attention (SILA), which incorporates a redesigned memory architecture and a weighted loss training strategy to encourage selective memory writing. SILA exhibits remarkable long-context retrieval capabilities, achieving $20\times$ context length extrapolation on the Passkey Retrieval task, and demonstrating superior memory utilization efficiency on the Needle-in-a-Haystack benchmark.

## 1 Introduction

The Transformer architecture and attention mechanism (Vaswani et al., 2017) have been the dominant architecture for language modeling over the past years. Transformer memorizes all past tokens in the form of KV cache for next-token generation, which makes it accurate even on long sequences, while also introducing the main drawback of the attention mechanism. As the sequence length grows, the size of KV cache grows linearly and the computation cost grows quadratically, which becomes a major bottleneck for efficient long-context inference.

Linear attention mechanisms are proposed to reduce the cost. Linear attention architectures are essentially RNNs (Katharopoulos et al., 2020)—the memory occupation remains constant on context with different length, and the computation cost grows linearly with sequence length. Despite their efficiency, linear attention architectures suffer from a significant drawback: they can only utilize constant memory space even when processing long contexts, placing them at a disadvantage in long-context tasks (Arora et al., 2024b;a; Bick et al., 2025).

However, most long-context tasks do not require memorizing the entire sequence to complete. Take the classic task of Needle-in-a-Haystack (NIAH) as an example (Hsieh et al., 2024):

> *A special magic number is hidden within the following text. Make sure to memorize it. I will quiz you about the number afterwards. ...(unrelated text)... The special magic number for tested-formal is: 3136088. ...(unrelated text)... What is the special magic number for tested-formal mentioned in the provided text?*

If humans are asked to perform such a task, the strategy they would adopt is: remember the initial instruction ("find the magic number") and then **ignore** all irrelevant text in the subsequent massive text until the target keywords ("magic number") appear. This strategy only requires constant memory overhead, regardless of sequence length. For linear attention models, this strategy means that much of the information in a sequence is neither forgotten after being written to memory, nor stored in a larger inference-time memory, but rather **never written into memory at all**. This strategy also

applies to a wider range of real-world long-text tasks, where an instruction is typically provided. The instruction serves as a clue, allowing the model to skip irrelevant content and focus on task-relevant segments, thus completing long-context tasks even when memory is strictly limited.

Consequently, we propose that enabling linear attention models to learn **selective ignoring** provides a promising approach to addressing long-context retrieval tasks under fixed memory capacity. Based on this principle, we make the following contributions:

- We re-evaluated the retrieval capabilities of popular linear attention models and observed the pattern of memory writing in these models. We found that these models complete the NIAH task through a specific preference for memorizing digit tokens, rather than demonstrating a general memorize-and-retrieve capability for arbitrary tokens. By redesigning the benchmark and intervention of memory writing, we explained how these models achieve inflated performance on the NIAH task and revealed their true retrieval capabilities.

- We propose Selective Ignoring Linear Attention (SILA), redesigning both the architecture of linear attention and its training strategy. We decouple the memory store and recall, and introduce a memory-dependent gate to address the observed memory writing preference. We identify and experimentally validate the conflict between standard next-token prediction training and selective ignoring, developing a weighted loss training strategy that implements differential weighting across tokens. Models with these enhancements demonstrate remarkable long-context retrieval capabilities.

## 2 BACKGROUND

**Linear Attention Models.** Different from Transformers, linear attention models use a memory $\mathcal{M}$ with constant capacity for sequence modeling. Generally, the update and readout of $\mathcal{M}$ can be written as online gradient descent (Behrouz et al., 2024):

$$\mathcal{M}_t = \gamma_t \mathcal{M}_{t-1} - \beta_t \nabla_{\mathcal{M}_{t-1}} \mathcal{L}(\mathcal{M}_{t-1}, \mathbf{k}_t, \mathbf{v}_t), \qquad \mathbf{o}_t = \mathcal{M}_t(\mathbf{q}_t) \tag{1}$$

where $\gamma_t$ and $\beta_t$ are input-dependent forget gate and input gate respectively, $\mathcal{M}$ is a differentiable parametric function, typically a linear layer. Appendix A offers a more detailed introduction to linear attention models. Variants of linear attention models (Table 5) include different design for forget gate (Yang et al., 2024b; 2025b), loss function (Behrouz et al., 2025a; von Oswald et al., 2025), structure of $\mathcal{M}$ (Sun et al., 2025; Behrouz et al., 2025a;b; 2024), layer architecture (Peng et al., 2025; Beck et al., 2025), optimizer for online SGD (Behrouz et al., 2025a) etc. Recent researches have proposed several hypotheses and enhancements for length extrapolation of linear attention models, such as the unexplored states hypothesis (Ruiz & Gu, 2025), limited effective receptive field hypothesis (Ben-Kish et al., 2025; Ye et al., 2025), and state overparameterization (Chen et al., 2024a). We offer a detailed analysis and comparison for these works in Appendix B.

**In-Context Retrieval.** The basic form of in-context retrieval, also referred to as in-context associative recall, are described as follows:

$$\underbrace{\dots \quad [A] \quad [B] \quad \dots}_{\text{context}} \quad [A] \quad \rightarrow \quad [B]$$

A **key** $[A]$ and an associative **value** $[B]$ is provided in the context. In the end of input, the model receives a **query** $[A]$, and it is expected to retrieve the associated value $[B]$. The tokens $[A]$ and $[B]$ can be arbitrary, so the key-value mapping can only be inferred from context instead of training data. Theoretically, vanilla attention is proven to solve in-context retrieval of arbitrary length, while linear attention models, do not have the guarantee (Arora et al., 2024a). Essentially, it's not possible to compress an infinite long sequence into a constant memory.

In real-world benchmarks, instructions are added before the context, so models have some information of $[A]$ and $[B]$ when searching the context. The instruction substantially impacts linear attention models as they process context unidirectionally, which is validated in (Arora et al., 2024b). Thus, to accurately evaluate the retrieval ability of linear attention models, we formalize the in-context retrieval task in this paper as follows:

$$\underset{\text{instruction}}{[A]} \quad \underbrace{\dots \quad [A] \quad [B] \quad \dots}_{\text{context}} \quad [A] \quad \rightarrow \quad [B]$$

By prepending an instruction containing information for $[A]$, this task becomes theoretically solvable with constant memory. This modification also aligns with the human cognitive patterns for such tasks, as described in Section 1.

# 3 RE-EVALUATING THE RETRIEVAL CAPABILITIES OF LINEAR ATTENTION MODELS

In this section, we propose several modifications to the original NIAH benchmark (Hsieh et al., 2024) based on some key observations, which allow us to better reveal the true long-context retrieval capabilities of linear attention models. Specifically, our analysis reveals that:

- The original NIAH benchmark evaluates models unreliably, leading to scores that are not comparable across different benchmarking frameworks.
- Linear attention models rely heavily on memorizing specific digits to achieve high performance on original NIAH tasks, which does not generalize to arbitrary retrieval targets.

## 3.1 BASIC SETUP

To evaluate the retrieval ability of linear attention models under instruction guidance in detail, we first propose two prompt variants of NIAH tasks:

1. No-instruction variant (*no inst* for short): no instruction is given before context. The model knows nothing about the task when processing the context, so that it must memorize the whole context to complete the task.

2. Strong-instruction variant (*strong inst* for short): the original NIAH instruction, together with the key for retrieval, is given before context, which the model can utilize to skip most of the context during reading.

Detailed prompt format can be found in Appendix C.1. For a basic comparison, Figure 1a shows the performance of linear attention models with *no inst* versus *strong inst*. Models show a performance improvement with *strong inst*, which is expected (Arora et al., 2024b). However, the improvement is marginal, and these models still fail to extrapolate effectively to longer sequences.

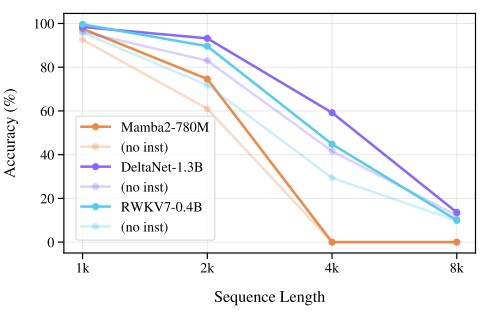
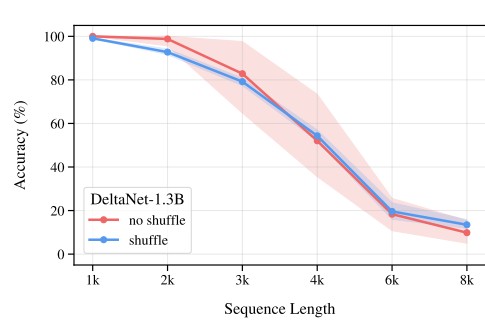

(a) NIAH-2 with *strong inst* and *no inst*          (b) NIAH-2 with(out) sample-level shuffling

Figure 1: (a) Performance of linear attention models on NIAH-2 with *strong inst* and *no inst*. The models show a slightly performance gain from the instruction. (b) Error range of measured scores with and without sample-level shuffling (with *strong inst*). Scores without sample-level shuffling are highly unstable, leading to inconsistent measurements across benchmark frameworks.

## 3.2 SAMPLE-LEVEL SHUFFLING OF HAYSTACKS

Secondly, we suggest shuffling the haystack for every sample during evaluation. Popular benchmark frameworks like RULER (Hsieh et al., 2024) and `lm-eval-harness` (Gao et al., 2024) use a same haystack across all samples. However, we observed that the success of retrieval is largely affected by the properties of the haystack. Since different benchmark frameworks use different

haystacks, scores are neither directly comparable nor stable. As a quantitative measure, we compared two settings: one using a single, fixed haystack for all samples (mirroring the original NIAH benchmark's behavior), and another using sample-level shuffling. We ran both benchmarks multiple times to measure their error range. As shown in Figure 1b, the benchmark without sample-level shuffling is highly unreliable, whereas sample-level shuffling makes the scores much more stable.

## 3.3 Preference for Digit Tokens in Memory Writing

Finally, we develop a variant of the NIAH benchmark that uses English words as retrieval targets instead of digits. This modification stems from a key experimental observation: linear attention models have a specific preference for memorizing digit tokens via special memory-writing patterns.

As shown in Equation 1, mainstream linear attention models control memory writing through an input gate $\beta_t$. Therefore, the variation of $\beta_t$ within a sequence indicates the model's preference for memorizing certain tokens. We found that some specialized memory heads respond actively (i.e., produce a high $\beta_t$) at the positions of digit tokens (i.e., 0-9) but very passively to most other tokens (Fig 2). In other words, these heads have a preference for memorizing digit tokens. In the original NIAH tasks, the value to be retrieved is always a digit string (decimal or hexadecimal). Therefore, the question arises: *Do these models complete the NIAH tasks using a general memorize-and-retrieve ability for arbitrary tokens, or do they rely on a specialized shortcut for digits to simplify the task?* To answer this question, we conducted two sets of experiments:

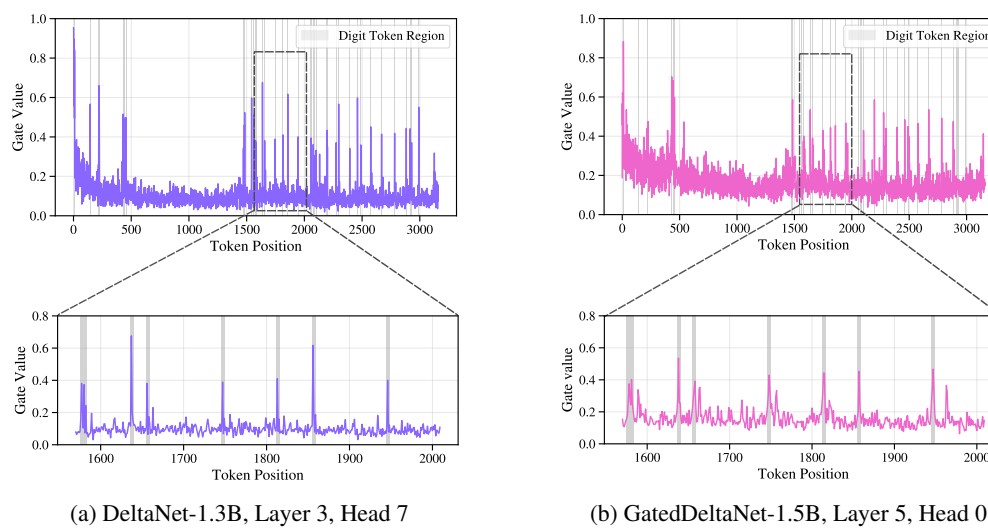

(a) DeltaNet-1.3B, Layer 3, Head 7        (b) GatedDeltaNet-1.5B, Layer 5, Head 0

Figure 2: Input gating patterns of DeltaNet-1.3B (left) and GatedDeltaNet-1.5B (right) in specific heads, which exhibit spiky patterns around digit tokens.

**The NIAH-Word Task**  We developed the NIAH-Word task, where the retrieval values are changed to an English phrase. We then re-evaluated the linear attention models on this new task and measured the performance gap compared to the case where retrieval values are digits (under

Table 1: NIAH (*no inst*) performance drop after changing the retrieval type from number to word.

| Model | NIAH-2 | | NIAH-Word | |
|---|---|---|---|---|
| | 1k | 2k | 1k | 2k |
| Qwen3-0.6B | 98.0 | 98.2 | 98.8 | 96.6 |
| Mamba2-370M | 92.6 | 61.0 | 53.0 (↓39.6) | 28.4 (↓32.6) |
| DeltaNet-1.3B | 96.2 | 83.0 | 25.8 (↓70.4) | 15.6 (↓67.4) |
| GatedDeltaNet-1.5B | 97.4 | 94.4 | 51.4 (↓46.0) | 29.6 (↓64.8) |
| RWKV7-0.4B | 95.8 | 71.8 | 37.2 (↓58.6) | 18.0 (↓53.8) |

*no inst* so that we can eliminate the effect from instructions). As shown in Table 1, all linear attention models suffer a significant performance drop on NIAH-Word, while a Transformer-based model (Yang et al., 2025a) is only slightly affected. This performance drop across different retrieval value types is sufficient to show that the claimed scores of linear attention models on NIAH benchmarks are not reliable: their performance is highly dependent on the retrieval value type, which is not generalizable.

**Intervention with Memory Writing**   In a subset of models, we recorded the positions of digit tokens in the input sample and overwrote the input gate value $\beta_t$ at these positions with the sequence's average gate value. We validated that this intervention is not destructive, as shown in Appendix C.2. This intervention resets the writing strength for digit tokens to the average level, therefore eliminating specialized writing strategy for them. After the intervention, retrieval scores dropped significantly (Fig 3). The results are sufficient to illustrate that the performance drop from the original NIAH task to the NIAH-Word task is primarily caused by specialized input gating strategies. As a gap still exists between NIAH with digit intervention and NIAH-Word, we hypothesize that other components(e.g. forget gates, MLPs) in linear attention models may also be more sensitive to digit tokens, since successful in-context retrieval requires not only storing tokens in memory (controlled by the input gate), but also retrieving the desired tokens from it.

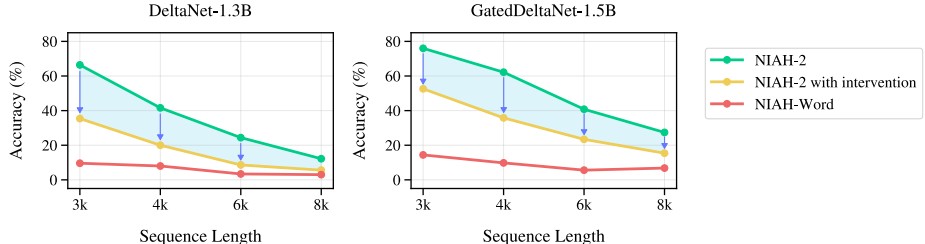

Figure 3: Effect of input gate intervention on NIAH (*no inst*) across sequence lengths in DeltaNet-1.3B (left) and GatedDeltaNet-1.5B (right).

The experiments above lead us to the conclusion that the general in-context retrieval ability of most linear attention models is not as powerful as their performance on the original NIAH tasks might suggest. Therefore, we suggest **NIAH-Word should be a necessary measurement for retrieval ability of linear attention models**. We also show how memory-writing patterns can help explain and even intervene with specific behaviors of linear attention models. This is also the inspiration for our architecture design.

## 4  MODEL ARCHITECTURE OF SILA

In this section, we reformulate the linear attention architecture following the principle of selective ignoring. To achieve enhanced selective memory writing capability, we postulate that the model must fulfill two core requirements:

1. Not every token is required to be written to memory;
2. The model can dynamically determine whether a new token should be stored based on the existing memory states.

**Decoupling of Memory Store and Recall**   Analysis of attention weights in pretrained Transformer models indicates that a substantial number of tokens focus only on themselves, recent tokens, and attention sinks (Xiao et al., 2025). Within the linear attention framework, the attention-weighted sum is replaced by the operation $\mathcal{M}_t(\mathbf{q}_t)$. To replicate this functionality, the memory state $\mathcal{M}_t$ must contain information corresponding to the current token, recent tokens, and attention sinks. Attention sinks can be achieved by a non-zero initialized memory state. However, the necessity to contain the current and recent tokens in $\mathcal{M}_t$ implies that they are always written into memory, whether it is truly important to remember or not. This leads to a tight coupling between memory writing (store) and reading (recall). Crucially, even accessing only the current token requires it to be first written into

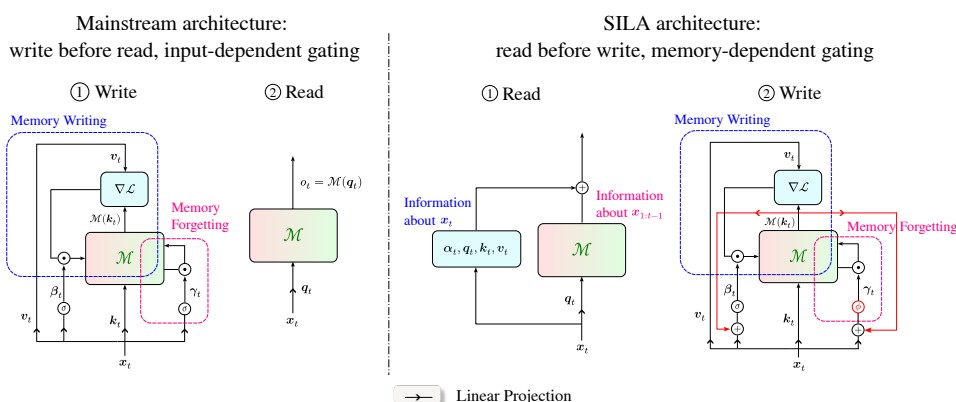

Figure 4: The illustration of mainstream linear attention architecture (left) and SILA (right).

memory. To address this limitation, we propose decoupling these operations: recall should depend solely on the past memory state while the current token is explicitly computed:

$$\mathbf{o}_t = \mathcal{M}_{t-1}(\mathbf{q}_t) + \alpha_t(\mathbf{k}_t^\top \cdot \mathbf{q}_t)\mathbf{v}_t \tag{2}$$

where $\alpha_t$ is a weight scalar. Since recall relies solely on $\mathcal{M}_{t-1}$, the decision of whether to store the current key-value pair $(\mathbf{k}_t, \mathbf{v}_t)$ into the memory state $\mathcal{M}_t$ is now completely decoupled. This directly fulfills requirement 1. Additionally, the use of short causal convolution (Yang et al., 2024b; 2025b; Allen-Zhu, 2025) enables $\mathbf{v}_t$ to directly incorporate neighboring token information, allowing tokens without long-range dependencies to be entirely omitted from memory storage. A similar implementation exists in RWKV7, termed "bonus". However, it still uses the updated state for recall, thus failed to achieve full decoupling.

**Memory-dependent Gate**  To fulfill requirement 2, a straightforward approach is to employ memory-dependent gates. The input gate and the forget gate are computed with not only the input $\mathbf{x}_t$, but also the information retrieved from memory using $\mathbf{k}_t$:

$$\square_t = \sigma(W_1 \mathcal{M}_{t-1}(\mathbf{k}_t) + W_2 \mathbf{x}_t), \square \in \{\beta, \gamma\} \tag{3}$$

To mitigate the continuous decay induced by the forget gate over long distances, we replace the sigmoid function with one that can reach 1 (ensuring no decay):

$$\phi(x) = \begin{cases} \frac{2}{e^x + e^{-x}} & x < 0 \\ 1 & x \geq 0 \end{cases} \tag{4}$$

Figure 4 illustrates our architecture design. SILA is broadly adapted from Gated DeltaNet (Yang et al., 2025b), with enhancements to the gate computations and memory recall as outlined above:

$$\beta_t = \text{sigmoid}(W_{\beta 1} \mathcal{M}_{t-1} \mathbf{k}_t + W_{\beta 2} \mathbf{x}_t), \quad \gamma_t = \phi(W_{\gamma 1} \mathcal{M}_{t-1} \mathbf{k}_t + W_{\gamma 2} \mathbf{x}_t) \tag{5}$$

$$\mathbf{o}_t = \mathcal{M}_{t-1} \mathbf{q}_t + \alpha_t(\mathbf{k}_t^\top \cdot \mathbf{q}_t)\mathbf{v}_t, \quad \alpha_t = \text{sigmoid}(W_\alpha \mathbf{x}_t) \tag{6}$$

$$\mathcal{M}_t = \gamma_t \mathcal{M}_{t-1} - \beta_t(\mathcal{M}_{t-1} \mathbf{k}_t - \mathbf{v}_t)\mathbf{k}_t^\top \tag{7}$$

## 5 TRAINING STRATEGY OF SILA

Standard next-token prediction training treats every token equally, with the final loss computed as the average of all token prediction losses. This approach works well for training Transformers, as they maintain full access to all previous tokens and can thus gradually optimize the prediction for each token. However, when applied to linear attention models, fixed memory capacity necessitates trade-offs: achieving accurate predictions for certain tokens within long contexts may inevitably compromise the prediction accuracy of others. Prior studies suggest that for a large portion of tokens, prediction is inherently easy and doesn't require reasoning or retrieval (Lin et al., 2024). Uniformly weighting the loss across all tokens may hinder the model's ability to fully develop reasoning and retrieval capabilities. Thus, this conventional training strategy conflicts with the goal of encouraging the model to learn selective ignoring.

## 5.1 INVESTIGATION OF TRAINING STRATEGIES ON SYNTHETIC RETRIEVAL TASK

To validate our hypothesis, we conducted experiments on a small-scale synthetic benchmark. We adapt a retrieval task (Fig 5) based on the in-context recall task from MAD-Lab (Poli et al., 2024). Following Section 2, we prefixed the input sequence with the retrieval key and employed a large vocabulary size to prevent the model from memorizing all possible key-value pairs. We evaluate two training strategies:

- Standard next-token prediction: compute loss over all tokens (Standard Loss).
- Target-only prediction: compute loss only on the final predicted answer (Target-only Loss).

We trained shallow 2-layer models (dim=64, num_heads=2, state_size=5248) with the sequence length of 128 and performed zero-shot evaluations on longer sequences. Models trained with Standard Loss (Fig 6a) exhibit a rapid decline in retrieval accuracy as the sequence length increases. Notably, our models trained with Standard Loss displayed substantial variability across runs (attributable to random initialization), suggesting the architecture may occasionally learn selective ignoring. In stark contrast, models trained with Target-only Loss (Fig 6b) demonstrated significantly superior length extrapolation performance. The performance is maintained even at sequence lengths 100 times greater than those encountered during training. The experimental results validate our hypothesis: to enable linear attention models to learn selective ignoring, it is imperative to avoid treating all tokens equally during training.

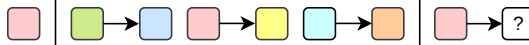

Figure 5: Synthetic retrieval task. Given a target key hint, the model needs to retrieve the corresponding value from an input sequence consisting of random key-value pairs.

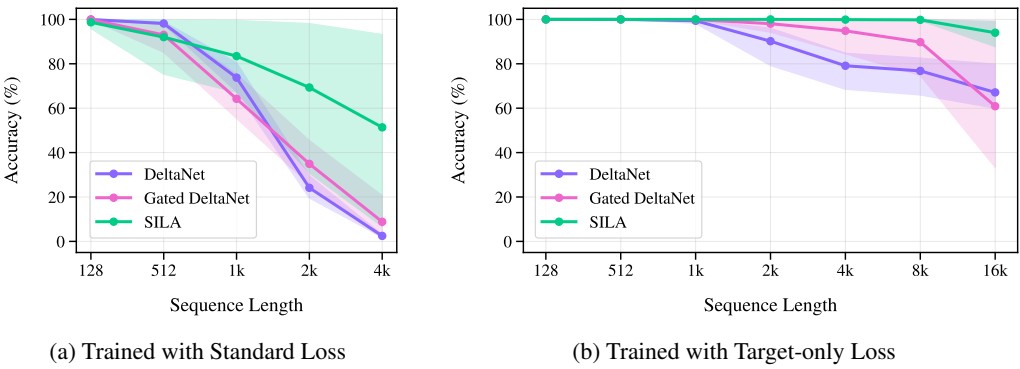

(a) Trained with Standard Loss  (b) Trained with Target-only Loss

Figure 6: Zero-shot evaluations on synthetic retrieval task.

## 5.2 TRAINING WITH WEIGHTED LOSS

We suggest that training linear attention models requires applying distinct weights to the loss of each token (weighted loss). To achieve this, we employ a simple yet effective approach: compute the weights using a reference pretrained Transformer model (Fig 7). This approach imposes no constraints on the architecture or size of the reference model, provided it possesses robust long-context retrieval capabilities. We first filter the attention weights, retaining only values above a threshold, and zero out the attention sinks (the first column). This allows us to approximate the most attended parts of each token. The filtered attention weights are then multiplied by the relative positional distance and summed to compute the average retrieval distance of each token. Finally, we apply logarithmic scaling to the average distances to derive the final weight for each token. This approach achieves our objective: tokens relying solely on local context have low weights, while tokens exhibiting long-distance dependencies have high weights. We also add an exponentially decaying constant term to the weights: $\exp(-training\_step/\tau) + weights$. This ensures that during the early stages of training, the model establishes a stable foundation in next-token prediction

capabilities, while progressively transitioning to learn selective ignoring behavior in later stages. Detailed pseudocode of loss weights computation is provided in Appendix C.3.

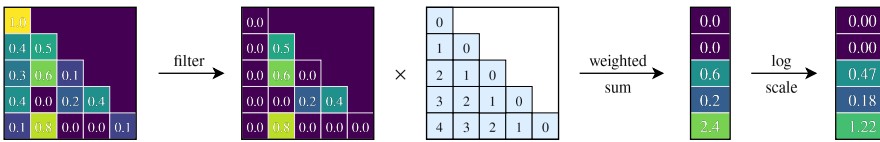

Figure 7: Computing token-level loss weights via filtered attention and distance-aware scaling.

# 6 EXPERIMENTS

We evaluate SILA on typical long-context retrieval tasks. Due to limited computational resources, we employed a small-scale 0.6B model (SILA-0.6B), with weights transferred from Qwen3-0.6B (see Appendix C.3 for details). Our model is trained on 15B tokens sampled from the FineWeb-Edu dataset (Penedo et al., 2024). The first 10B tokens are trained with a context length of 1024, and the remaining 5B tokens are trained with a context length of 4096. To validate the generality of the proposed training strategy and compare architectural differences, we also trained a Gated DeltaNet under identical configurations (GatedDelta+Ours-0.6B). All other baselines are from open-source pretrained models (see Appendix C.4 for details). Although variations in model scale, training data, and total training tokens may introduce some bias in the evaluations, the results remain sufficient to substantiate our conclusions.

**Passkey Retrieval** Figure 8 shows the results of the Passkey Retrieval task (Chen et al., 2024b), which requires the model to retrieve a random number embedded within a long document of repeated sentences. Retrieval tends to be more challenging when the number is inserted near the beginning of the document (corresponding to a lower passkey depth), as it is farther from the final query. Surprisingly, SILA maintains high accuracy until around **80k** tokens. This is particularly notable given that our model is only trained on a maximum context length of **4k** without any fine-tuning.

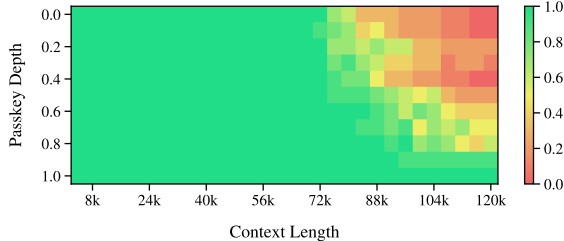

Figure 8: Results on Passkey Retrieval task.

**Needle in a Haystack** Following the NIAH benchmark outlined in Section 3, we evaluated SILA and popular linear attention models. As shown in Table 2, SILA exhibits the least performance degradation over long context lengths. Meanwhile, Gated DeltaNet trained with our proposed training strategy also shows outstanding performance. Given the differences in parameter counts and state sizes between models, we further visualize the state size vs. NIAH performance landscape in Figure 9. The results reveal that SILA achieves a substantial lead in memory utilization efficiency.

**Commonsense reasoning** An intuitive concern is whether our training strategy compromises general task performance. To address this, we evaluated models on commonsense reasoning benchmarks (Table 3). Compared to standard loss (i.e. *w/o. weighted loss*), the model trained with weighted loss exhibits slight deterioration on certain tasks. Notably, prior studies have indicated that such performance differences fall well within typical error margins (3%~5%) attributable to random training seeds (Allen-Zhu, 2025) and are often less significant than variations induced by prompt engineering. Overall, all models exhibit comparable performance on commonsense reasoning tasks, as scaling laws suggest that such capabilities are primarily determined by training data composition and total training tokens (Chang et al., 2024; Grattafiori et al., 2024).

Table 2: Performance comparison on NIAH (*strong inst*) benchmark.

| Model | State Size | NIAH-1 | | | | NIAH-2 | | | NIAH-Word | | |
|---|---|---|---|---|---|---|---|---|---|---|---|
| | | 8k | 16k | 24k | 32k | 2k | 4k | 8k | 1k | 2k | 4k |
| Qwen3-0.6B | 57344×seqlen | 100.0 | 100.0 | 100.0 | 100.0 | 100.0 | 100.0 | 100.0 | 98.6 | 97.0 | 94.0 |
| Mamba2-370M | 13025280 | 100.0 | 54.4 | 24.8 | 9.0 | 73.2 | 14.0 | 1.4 | 86.2 | 32.2 | 2.4 |
| Mamba2-780M | 19513344 | 100.0 | 0.2 | 0.0 | 0.0 | 74.6 | 0.0 | 0.0 | 82.2 | 37.4 | 0.0 |
| DeltaNet-1.3B | 6881280 | 100.0 | 100.0 | 100.0 | 100.0 | 93.2 | 59.2 | 13.6 | 47.4 | 20.4 | 6.8 |
| Gated DeltaNet-1.5B | 13172736 | 94.8 | 62.0 | 38.8 | 33.6 | 97.0 | 74.2 | 30.6 | 74.0 | 37.8 | 11.6 |
| RWKV7-0.4B | 1622016 | 100.0 | 99.0 | 62.6 | 10.4 | 89.6 | 44.8 | 10.0 | 57.8 | 26.2 | 10.2 |
| RWKV7-1.5B | 3244032 | 100.0 | 100.0 | 99.4 | 32.4 | 99.0 | 82.6 | 20.2 | 82.6 | 67.4 | 33.2 |
| GatedDelta+Ours-0.6B | 2179072 | 100.0 | 100.0 | 99.6 | 85.4 | 96.0 | 82.4 | 28.6 | 73.0 | 43.2 | 15.0 |
| *w/o. weighted loss* | 2179072 | 83.2 | 39.8 | 19.6 | 11.8 | 54.8 | 16.8 | 7.2 | 24.2 | 9.6 | 3.0 |
| SILA-0.6B | 2179072 | 100.0 | 100.0 | 100.0 | 100.0 | 98.4 | 90.2 | 49.2 | 85.0 | 63.6 | 25.8 |
| *w/o. weighted loss* | 2179072 | 76.0 | 40.0 | 27.4 | 13.2 | 71.2 | 28.8 | 9.8 | 43.8 | 15.6 | 4.8 |

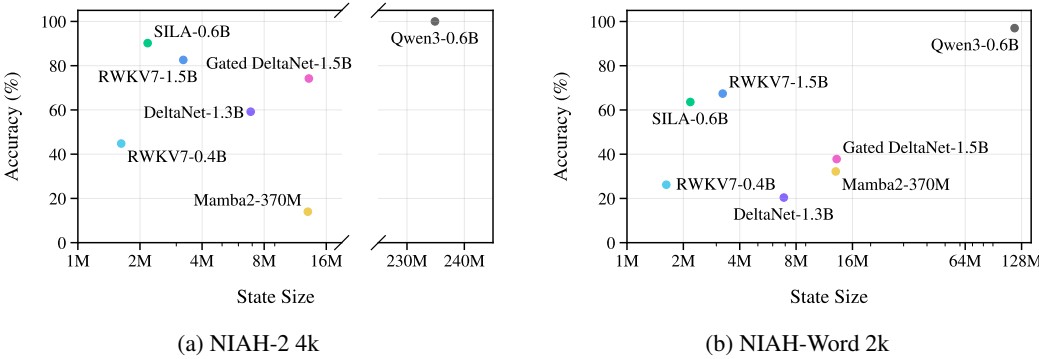

(a) NIAH-2 4k          (b) NIAH-Word 2k

Figure 9: State size vs. accuracy on NIAH benchmarks.

Table 3: Zero-shot performance comparison on commonsense reasoning.

| Model | # Training Tokens(B) | ARC-e acc | ARC-c acc | LMB. acc | Hella. acc_n | Wino. acc | PIQA acc | Avg. |
|---|---|---|---|---|---|---|---|---|
| Qwen3-0.6B | 30000 | 65.74 | 33.45 | 53.77 | 53.79 | 58.88 | 69.80 | 55.91 |
| Mamba2-370M | 300 | 54.92 | 25.09 | 55.79 | 46.92 | 55.33 | 70.46 | 51.42 |
| Mamba2-780M | 300 | 61.03 | 26.71 | 61.52 | 54.92 | 60.06 | 72.09 | 56.06 |
| DeltaNet-1.3B | 100 | 58.59 | 24.49 | 48.36 | 50.22 | 52.80 | 70.62 | 50.85 |
| Gated DeltaNet-0.4B | 20 | 60.27 | 25.68 | 34.72 | 41.48 | 50.43 | 66.05 | 46.44 |
| RWKV7-0.4B | 2000 | 68.22 | 31.74 | 58.76 | 56.72 | 59.98 | 72.47 | 57.98 |
| GatedDelta+Ours-0.6B | 15 | 61.57 | 30.12 | 45.31 | 47.29 | 54.06 | 67.74 | 51.02 |
| *w/o. weighted loss* | 15 | 67.89 | 34.30 | 41.22 | 48.01 | 56.12 | 68.88 | 52.74 |
| SILA-0.6B | 15 | 65.45 | 31.91 | 42.67 | 44.40 | 55.33 | 67.25 | 51.17 |
| *w/o. weighted loss* | 15 | 66.96 | 33.02 | 41.29 | 48.44 | 56.27 | 69.15 | 52.52 |

**Ablation & Analysis** We designed experiments to separately evaluate the impacts of model architecture and training strategy. As shown in Table 2, the most significant improvement stems from our proposed training strategy. For both SILA-0.6B and GatedDelta+Ours-0.6B, replacing the standard loss with our weighted loss leads to substantial performance gains. In addition, the performance difference between GatedDelta+Ours-0.6B and SILA-0.6B can also exhibit the improvements attributable to architectural design. Notably, these results suggest that our models could have benefited from more extensive training. GatedDelta+Ours-0.6B without the weighted loss (which corresponds to a standard Gated DeltaNet initialized from Qwen3-0.6B and trained on 15B tokens) exhibits a gap compared to the well-pretrained baseline models, indicating insufficient training. Nevertheless, SILA-0.6B outperforms these baselines, despite their larger training data and parameter sizes.

To verify whether our model can leverage instructions to achieve selective ignoring and thereby enhance retrieval performance, we evaluated models under *no inst* and *strong inst* settings on NIAH-Word. Results in Table 4 show that SILA-0.6B obtains a clear performance boost from instructions, while other models exhibit only small improvements. We also provide a visualization of memory writing patterns in Appendix D, which explains how the model uses selective memory writing to enhance retrieval performance.

Table 4: Performance of linear attention models on NIAH-Word with *no inst* and *strong inst*. SILA-0.6B shows a significant performance gain from the instruction.

| Model | *no inst* | | *strong inst* | |
|---|---|---|---|---|
| | 2k | 4k | 2k | 4k |
| Mamba2-780M | 28.4 | 0.0 | 37.4 (↑9.0) | 0.0 (↑0.0) |
| DeltaNet-1.3B | 15.6 | 8.0 | 20.4 (↑4.8) | 6.8 (↑0.0) |
| GatedDeltaNet-1.5B | 29.6 | 9.8 | 37.8 (↑8.2) | 11.6 (↑1.8) |
| RWKV7-0.4B | 18.0 | 9.6 | 26.2 (↑8.2) | 10.2 (↑0.6) |
| SILA-0.6B | 31.0 | 10.6 | 63.6 (↑32.6) | 25.8 (↑15.2) |

## 7 CONCLUSION

In this work, we re-examine the long-context retrieval capabilities of linear attention models from a memory writing perspective. Following the principle of selective ignoring, we propose improvements to the model architecture and training methodology to enhance long-context performance.

Despite promising results, this work has limitations that open avenues for future research. While our implementation employs a straightforward training strategy dependent on pretrained Transformers, designing novel training paradigms to enable self-supervised learning for selective ignoring remains an open challenge. Furthermore, in this study, we still assume that the model can read the text only once to complete the task. Such a unidirectional pass is inherently limited compared to the full context access ability of a Transformer. Given the advantage of linear complexity, linear attention models with selective ignoring capabilities could potentially achieve both efficiency and accuracy when integrated with controlled look-back mechanisms.

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

## A INTRODUCTION TO LINEAR ATTENTION MODELS

In language modeling tasks, Transformers use the softmax attention mechanism:

$$Q = XW_Q, K = XW_K, V = XW_V, \qquad O = \text{softmax}(\frac{QK^\top}{\sqrt{d_k}})V. \tag{8}$$

where $W_Q, W_K, W_V \in \mathbb{R}^{d \times d}, X, Q, K, V, O \in \mathbb{R}^{T \times d}$ (consider a single attention head for simplicity). The quadratic complexity $\mathcal{O}(T^2)$ comes from the computation of attention map $\text{softmax}(QK^\top/\sqrt{d_k})$. However, if the attention map can be decoupled into $\phi(Q)\phi(K)$ (where $\phi$ is usually an element-wise nonlinear function like SiLU), we will get the original version of linear attention (Katharopoulos et al., 2020):

$$O = (\phi(Q)\phi(K)^\top)V = \phi(Q)(\phi(K)^\top V). \tag{9}$$

which has linear complexity $\mathcal{O}(T)$. Equation 9 can also be written as a recurrent form: it is mathematically equivalent to

$$\mathcal{M}_t = \mathcal{M}_{t-1} + \mathbf{v}_t \phi(\mathbf{k}_t)^\top, \quad \mathcal{M}_0 = 0 \tag{10}$$

$$\mathbf{o}_t = \mathcal{M}_t \phi(\mathbf{q}_t) \tag{11}$$

Update in equation 10 can also be seen as gradient descent with respect to $\mathcal{L} = -\mathbf{v}_t^\top \cdot (\mathcal{M}_{t-1}\phi(\mathbf{k}_t))$. More generally, most variants attention mechanism of linear attention models can be described as online gradient optimization on arbitrary memory $\mathcal{M}$ with constant capacity:

$$\mathcal{M}_t = \gamma_t \mathcal{M}_{t-1} - \beta_t \nabla_{\mathcal{M}_{t-1}} \mathcal{L}(\mathcal{M}_{t-1}, \mathbf{x}_t) \tag{12}$$

$$\mathbf{o}_t = \mathcal{M}_t(\mathbf{q}_t) \tag{13}$$

Some architectures do not obey Equation 1 strictly, but the paradigm is very similar (Beck et al., 2024). Some typical instances of linear attention mechanisms are listed in Table 5. $\gamma_t$ and $\beta_t$ are forget gate and input gate respectively, which is input-dependent by $\gamma_t = \sigma(W_\gamma \mathbf{x}_t)$, $\beta_t = \sigma(W_\beta \mathbf{x}_t)$. From the online gradient descent perspective, the $\gamma_t$ acts as weight decay factor, and the $\beta_t$ acts as the learning rate.

Table 5: Linear attention architectures.

| Model | Memory Update Rule |
|---|---|
| Mamba2 (Dao & Gu, 2024) | $\mathcal{M}_t = \gamma_t \mathcal{M}_{t-1} + \mathbf{v}_t \mathbf{k}_t^\top$ |
| DeltaNet (Yang et al., 2024b) | $\mathcal{M}_t = \mathcal{M}_{t-1}(\mathbf{I} - \beta_t \mathbf{k}_t \mathbf{k}_t^\top) + \beta_t \mathbf{v}_t \mathbf{k}_t^\top$ |
| Gated DeltaNet (Yang et al., 2025b) | $\mathcal{M}_t = \mathcal{M}_{t-1}\gamma_t(\mathbf{I} - \beta_t \mathbf{k}_t \mathbf{k}_t^\top) + \beta_t \mathbf{v}_t \mathbf{k}_t^\top$ |
| RWKV7 (Peng et al., 2025) | $\mathcal{M}_t = \mathcal{M}_{t-1}(\text{diag}(\gamma_t) - \beta_t \hat{\kappa}_t \hat{\kappa}_t^\top) + \beta_t \mathbf{v}_t \tilde{\mathbf{k}}_t^\top$ |
| MesaNet (von Oswald et al., 2025) | $H_t = \gamma_t H_{t-1} + \beta_t \mathbf{k}_t \mathbf{k}_t^\top,$ 
 $G_t = \gamma_t G_{t-1} + \beta_t \mathbf{v}_t \mathbf{k}_t^\top$ |
| TTT (Sun et al., 2025) | $\mathcal{M}_t = \mathcal{M}_{t-1} - \beta \nabla_{\mathcal{M}_{t-1}} \mathcal{L}(\mathcal{M}_{t-1}, \mathbf{k}_t, \mathbf{v}_t)$ |
| Titans (Behrouz et al., 2024) | $\mathcal{S}_t = \eta_t \mathcal{S}_{t-1} - \beta \nabla_{\mathcal{M}_{t-1}} \mathcal{L}(\mathcal{M}_{t-1}, \mathbf{k}_t, \mathbf{v}_t),$ 
 $\mathcal{M}_t = \gamma_t \mathcal{M}_{t-1} + \mathcal{S}_t$ |

## B ANALYSIS ON EXISTING EXTRAPOLATION METHODS

Several recent studies have already investigated length generalization, also known as length extrapolation, in linear attention models. Some works like DeciMamba (Ben-Kish et al., 2025) and LongMamba (Ye et al., 2025) also proposed to enhance long-context capability through skipping tokens during memory writing. Specifically, they manually suppress the writing strength of tokens deemed less important (e.g., by setting their update weights to zero when the writing strength is below a threshold or through top-k selection) in layers responsible for long-range dependencies. However, assuming the existence of bias for digit tokens (Section 3.3), merely suppressing writing strength may not theoretically enhance general retrieval capabilities. For validation, we measured

the performance of these methods under the benchmark framework proposed in Section 3. It turns out that these extrapolation methods indeed enhance performance on retrieval tasks related to digits (NIAH-2). However, on more general retrieval tasks (NIAH-Word), their performance is only comparable to, or even inferior to that of the base model. This observation aligns with our conclusion of digit preference.

Table 6: Performance of existing extrapolation methods compared to their base models. SP is abbreviation for state-passing.

| Model | NIAH-2 | | | | NIAH-Word | | |
|---|---|---|---|---|---|---|---|
| | 1k | 2k | 4k | 8k | 1k | 2k | 4k |
| Mamba-130M | 52.8 | 9.2 | 3.0 | 2.6 | 16.0 | 2.4 | 2.0 |
| DeciMamba-130M | 84.2 | 72.8 | 12.6 | 1.4 | 6.4 | 3.4 | 0.8 |
| Mamba2-1.3B | 96.4 | 61.8 | 1.0 | 0.0 | 46.8 | 12.4 | 0.0 |
| LongMamba-1.3B | 97.0 | 61.8 | 33.6 | 16.0 | 50.2 | 13.8 | 4.6 |
| Mamba2-370M | 98.4 | 73.2 | 14.0 | 1.4 | 86.2 | 32.2 | 2.4 |
| Mamba2-370M (SP) | 99.4 | 86.4 | 22.8 | 10.4 | 69.0 | 20.8 | 3.6 |
| SILA-0.6B | 99.6 | 98.4 | 90.2 | 49.2 | 85.0 | 63.6 | 25.8 |

Another popular approach to enhancing the long-context processing capability of linear attention models is state-passing (SP), or truncated backpropagation through time (TBTT). During training, SP initializes the initial states of each sequence segment with the final states of the preceding segment, thereby effectively simulating longer sequence length in training or post training. Prior studies have shown that applying SP to linear attention models, such as those in the Mamba family, can effectively mitigate the explosion of pointwise perplexity on long sequences (Yang et al., 2024a; Ruiz & Gu, 2025; Hu et al., 2025).

To validate the effectiveness of state-passing on long-context retrieval tasks, we conducted state-passing on Mamba2-370M official checkpoint with a setting similar to (Ruiz & Gu, 2025), and investigated the property of pointwise perplexity in various linear attention models. For state-passing, we concatenated input samples from FineWeb-Edu (Penedo et al., 2024) into sequences of 24k tokens and stopped gradients every 2k tokens. It turns out that state-passing significantly reduces pointwise perplexity on long sequences for Mamba2-370M (Fig 10), with improvement on NIAH-2 tasks (Table 6), but slight degradation on NIAH-Word tasks, suggesting that SP may not consistently improve general retrieval capabilities.

Notably, the perplexity explosion phenomenon appears to be specific to Mamba-style architectures and is not observed in other linear attention models we tested (Fig 10). This implies that the issue may stem from architectural characteristics rather than being a universal limitation of linear attention mechanisms.

## C EXPERIMENTAL DETAILS

### C.1 NIAH BENCHMARK SETUP

An example original NIAH prompt format is given as following:

> *A special magic number is hidden within the following text. Make sure to memorize it. I will quiz you about the number afterwards. ...(unrelated text)... One of the special magic numbers for tested-formal is: 3136088. ...(unrelated text)... What is the special magic number for tested-formal mentioned in the provided text?*

The instruction part does provide guidance about the target of retrieval, i.e. the target is a string of digits. The guidance can be weakened by providing no instruction at all, or be further strengthened by providing the key for retrieval. To clearly evaluate the effect of instruction guidance, we conducted evaluation of the weakened and strengthened variant in this research.

The *no inst* variant corresponds to the weakened version, e.g.

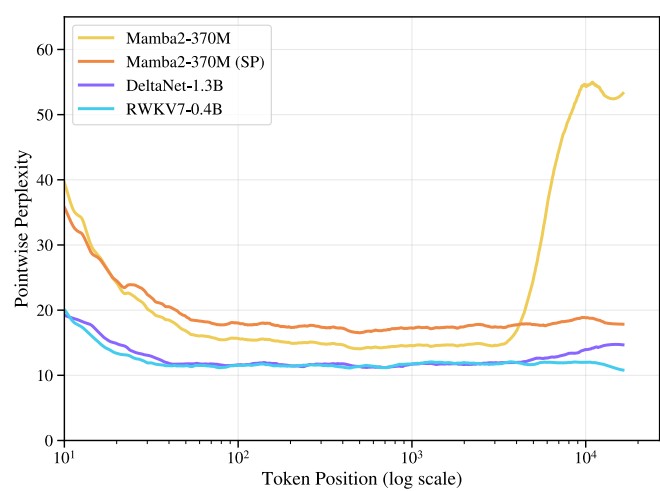

Figure 10: Pointwise perplexity of various linear attention models.

> *...(unrelated text)... One of the special magic numbers for tested-formal is: 3136088. ...(unrelated text)... What is the special magic number for tested-formal mentioned in the provided text?*

The *strong inst* variant corresponds to the strengthened version, e.g.

> *A special magic number is hidden within the following text. Make sure to memorize it. I will quiz you about the number for tested-formal afterwards. ...(unrelated text)... One of the special magic numbers for tested-formal is: 3136088. ...(unrelated text)... What is the special magic number for tested-formal mentioned in the provided text?*

Under the unidirectional reading paradigm of linear attention models, the *no inst* variant requires the model to memorize the whole context to give the answer in the end, while the *strong inst* variant provides the option to skip most of the context.

## C.2 VERIFICATION ON MEMORY WRITING INTERVENTION

In Section 3.3, we manually tweaked the writing strength in pretrained linear attention models. Here we verify that the intervention is not destructive on general language modeling capabilities.

To verify this, we evaluate the models on commonsense benchmarks, with 4% (average percentage of digit tokens in NIAH samples) token positions randomly chosen to reset the corresponding input gate value to the average in sequence. We conduct this intervention across all layers and heads of the model. We validate this setup on the LAMBADA benchmark. After random resetting of the input gate value, the accuracy of DeltaNet-1.3B dropped from 48.36 to 47.62 (↓0.74), and the accuracy of GatedDeltaNet-1.5B dropped from 50.16 to 49.16 (↓1.0), which we consider as marginal.

Therefore, resetting the writing strength under this percentage generally does not corrupt the capabilities of linear attention models. With observation that similar intervention strongly affects the performance of these models on NIAH benchmarks, we come to the conclusion that these models specially memorize the tokens on the affected positions and utilize them for prediction, as stated in Section 3.3.

## C.3 TRAINING DETAILS

To reduce training costs, we first initialize our model by copying the embedding and MLP layer weights from Qwen3, aligning the outputs of the linear attention layer with those of the Qwen3 attention layer on only 200M tokens. Specifically, we feed the hidden states from each Qwen3 layer

to the linear attention layer and minimize the MSE loss between its output and that of Qwen3's attention layer. This process requires minimal computational overhead but provides a strong weight initialization.

The pseudocode of loss weights computation is outlined in Algorithm 1. In subsequent pretraining, the loss weight for each predicted token is set to $\exp(-training\_tokens/10^9) + weights$. This allows the model to initially learn basic next-token prediction capability, while the training progressively transitions to pure weighted loss after around 5B tokens.

---

**Algorithm 1** Compute Token-Level Loss Weights

---

**Input:** Attention weights $\mathbf{A} \in \mathbb{R}^{L \times H \times T \times T}$ from a reference Transformer
**Hyperparameter:** Threshold $\tau$ (e.g., $\tau = 0.2$), Scaling factor $\lambda$ (e.g., $\lambda = 0.5$)
**Output:** Loss weights $\mathbf{w} \in \mathbb{R}^T$
Zero out the first column of $\mathbf{A}$                                        ▷ remove attention to sink token
$\mathbf{A} \leftarrow \mathbf{A} \cdot \mathbb{I}[\mathbf{A} \geq th]$           ▷ thresholding
**for** each token position $t = 1$ to $T$ **do**
    $w_t \leftarrow \frac{1}{LH} \sum_{l=1}^{L} \sum_{h=1}^{H} \sum_{j=1}^{t} A_{l,h,t,j} \cdot (t-j)$       ▷ compute average retrieval distance
    $w_t \leftarrow \log(\lambda \cdot w_t + 1)$                                  ▷ log scaling
**end for**
**return w**

---

Our models are trained on a total of 15B tokens sampled from the FineWeb-Edu dataset (Penedo et al., 2024): the first 10B tokens use a context length of 1024, while the remaining 5B tokens are trained with an extended context length of 4096. No further post-training or fine-tuning is performed.

### C.4 BASELINES

Due to constraints on computation resources, we employed existing pretrained models. Source of used pretrained checkpoints are listed in Table 7. For DeltaNet, Mamba2 and RWKV7 series, we used official checkpoints on HuggingFace. For GatedDeltaNet series, we used checkpoints from m-a-p since there are no official checkpoints.

For inference frameworks, we used `flash-linear-attention` for most models, and official implementation of `mamba-ssm` for Mamba2 series, custom Triton implementation for our architecture. It should be noticed that RWKV7 series also have official implementation in CUDA. As different implementations for inference of RWKV7 series show no difference on retrieval tasks, we used implementation in flash-linear-attention for better compatibility to evaluation benchmarks.

Table 7: Pretrained checkpoints used in this research with links on HuggingFace.

| Model Name | HuggingFace Checkpoint |
|---|---|
| Qwen3-0.6B | `Qwen/Qwen3-0.6B-Base` |
| Mamba-130M | `state-spaces/mamba-130m` |
| Mmaba2-370M | `state-spaces/mamba2-370m` |
| Mamba2-780M | `state-spaces/mamba2-780m` |
| DeltaNet-1.3B | `fla-hub/delta_net-1.3B-100B` |
| Gated DeltaNet-0.4B | `m-a-p/340M-20B-GatedDeltaNet-pure-baseline` |
| Gated DeltaNet-1.5B[*] | `m-a-p/1.3B-100B-GatedDeltaNet-pure` |
| RWKV7-0.4B | `fla-hub/rwkv7-0.4B-world` |
| RWKV7-1.5B | `fla-hub/rwkv7-1.5B-world` |

[*] The checkpoint `m-a-p/1.3B-100B-GatedDeltaNet-pure` has an actual parameter count of 1.5B.

As most of the tested models are not instruction-tuned and do not support chat template officially, we didn't apply any chat template during evaluation. However, it's worth noting that difference in template does influence the retrieval performance. We evaluated RWKV7 series both with and without chat template as it's offcially supported, and found that chat template generally improved the retrieval score (Table 8), although RWKV7 is declared to be trained without instruction tuning (Peng

et al., 2025). To exclude factors related to prompt engineering and instruction tuning, we reported all results in this research with unified template in completion style.

Table 8: Performance comparison w/ or w/o chat template. (*strong inst*)

| Model | NIAH-1 | | NIAH-2 | | | NIAH-Word | | |
|---|---|---|---|---|---|---|---|---|
| | 16k | 24k | 2k | 4k | 8k | 1k | 2k | 4k |
| RWKV7-0.4B(w/o template) | **99.0** | **62.6** | 89.6 | 44.8 | 10.0 | 57.8 | 26.2 | **10.2** |
| RWKV7-0.4B(w/ template) | 96.0 | 54.2 | **98.4** | **71.0** | **14.2** | **63.8** | **34.2** | 9.6 |

## D  VISUALIZATION OF MEMORY WRITING

To verify that SILA learns selective ignoring to improve long-context retrieval, we analyzed the input gate activity in all memory heads and layers of SILA-0.6B. We found some specific patterns as shown in Figure 11. In these heads, once the prompt is prefixed by instruction, they will respond significantly stronger at the retrieval key and answer tokens, while suppressing activity at other positions. By selectively enhancing and suppressing input gate values across the sequence, the model can achieve more effective memory management under instruction guidance.

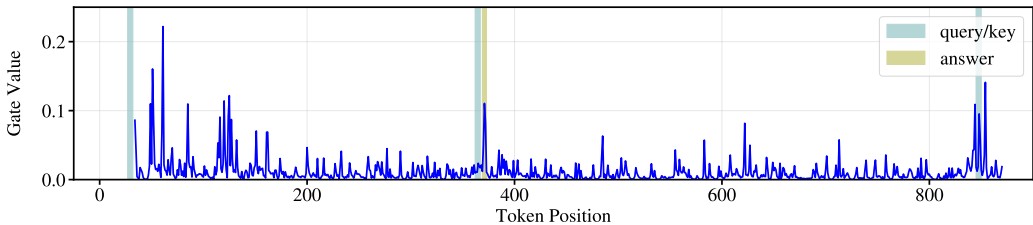

(a) Input gate patterns on *no inst* variant of NIAH-Word sample

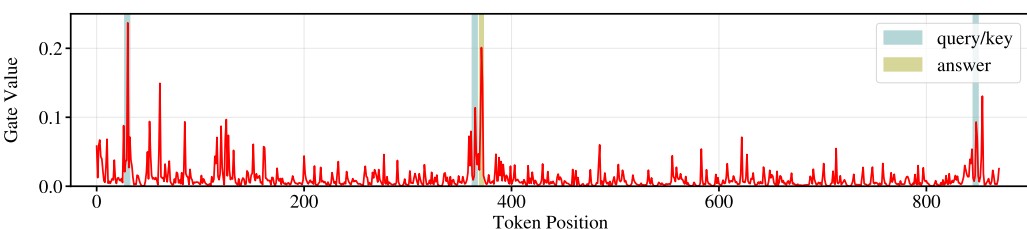

(b) Input gate patterns on *strong inst* variant of NIAH-Word sample

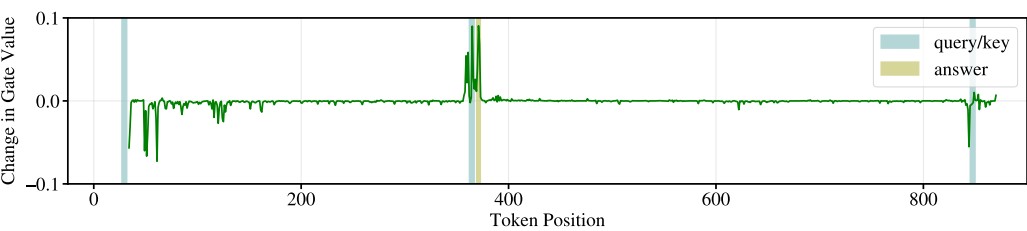

(c) Difference between (b) and (a)

Figure 11: The behavior of input gate in layer 22, head 14 in SILA-0.6B when processing one NIAH-Word sample with *no inst* and *strong inst* variant. The positions of query/key and answer are marked. Significant growth of gate value is only observed around retrieval key and answer tokens, while suppression is widely observed in other regions. Under *strong inst*, the input gate attains its highest values precisely at the query and answer positions, which is consistent with the intended behavior of selective ignoring.

