# OpenReview forum: "SILA: Enhancing Long-Context Retrieval Capability of Linear Attention via Selective Ignoring"
_ICLR.cc/2026/Conference — Submitted to ICLR 2026_

### Official Review · Reviewer_PcX2 · 2025-10-29

**Soundness:** 3
**Presentation:** 3
**Contribution:** 3
**Rating:** 4
**Confidence:** 4

**Summary:**

This paper introduces SILA, which enhances linear attention retrieval through improvements in both recurrence modeling and training objectives. Its core idea is to allow the model to selectively disregard irrelevant tokens.

The authors begin with an analysis of the NIAH task and establish a more reliable evaluation setting. Motivated by the need for selective memory writing, SILA incorporates several architectural innovations: it decouples the “read” and “write” operations (i.e., read-before-write) and introduces state-dependent gates. In addition, the authors employ a progressive selective weighted loss, based on a pretrained transformer, to improve the efficiency of long-context training.

Experimental results under the Transformer-to-RNN paradigm demonstrate that SILA improves the retrieval capability of linear attention models while maintaining competitive performance on general short-context modeling.

**Strengths:**

1. The motivation and proposed approach of this work are well-reasoned and coherent. The paper identifies a key deficiency of linear attention models under NIAH-style tasks and demonstrates that this issue can be alleviated through selective ignoring.

2. The analysis of the NIAH benchmark is logical and convincing, ensuring the reliability of the experimental evaluation.

3. The improvements in both modeling and training are essential and directly address the need for selective ignoring.

**Weaknesses:**

1. Some of the conclusions are not thoroughly or fairly validated. For example, the passkey retrieval extrapolation results of Qwen3 and GatedDelta are not shown; since the transferred Qwen3 checkpoints already support 32k context without ntk scaling, further clarification is needed here. In addition, in Table 5, only SILA is a transferred model, making it difficult to fairly compare the effectiveness of the proposed method with other selective writing approaches such as LongMamba.

2. The paper lacks ablation studies and detailed analyses for the proposed improvements. It remains unclear how each component—such as the training strategy and the two architectural modifications—specifically contributes to retrieval performance or NIAH results.

3. The experimental setting is overly constrained. The model scale (0.6B), training length (1k/4k), and evaluation scenario (Transformer-to-RNN transfer) are all highly specialized. Under such a narrow setup, it is difficult to disentangle how much of the observed gain arises from the proposed method itself rather than from others like parameter increases or tuning. Moreover, there are concerns regarding the scalability of the approach.

**Questions:**

1. In Table 1, the in-context retrieval capability appears correlated with training data scale. Since Qwen3 was trained on 36T high-quality tokens while other linear models were significantly undertrained, would a Transformer trained on a comparable corpus (e.g., hundreds of billions of tokens) still demonstrate similarly robust NIAH-Word performance?

2. Compared to linear attention baselines such as GatedDeltaNet, how do SILA’s computational and parameter costs change? What are the shapes of $W_{\gamma}$ and $W_{\beta}$?

3. What is the rationale for the chosen functional form of the forget gate $\phi(x)$ on the negative axis in Eq.10, which seems relatively uncommon?

4. As the memory-dependent gates introduce nonlinearity in the recurrence via the hidden state, does this hurt parallelism or training efficiency, particularly for long-context sequences? From an implementation standpoint, is it necessary to materialize all intermediate states across time steps?

5. Regarding the weighted loss, how were the threshold and scaling factor determined? Were these hyperparameters tuned through systematic sweeps?

6. What are the NIAH results for SILA (standard loss)? Only commonsense reasoning results are reported.

7. To support the claim that the improvement stems from selective ignoring, it would be helpful to visualize and analyze the gate values (${\gamma}_t$ and ${\beta}_t$) and compare them against those of GatedDeltaNet.

---

> ### Author Response · Authors · 2025-11-23
>
> Thank you for your valuable feedback. We have revised the manuscript accordingly and address each of your points as follows:
> > W1:
> 1) NIAH-1 task is the passkey retrieval task with only minor prompt variations. Therefore, the NIAH-1 results in Table 2 directly address your concern for passkey retrieval comparison.
> 2) We would like to clarify that the long-context capability of Qwen3 is not able to be transferred to linear attention models through weight initialization. This is validated by our experiment in Table 2: a Gated DeltaNet initialized from Qwen3-0.6B and trained with the standard loss (GatedDelta+Ours-0.6B w/o weighted loss) shows poor performance on NIAH. Thus, the fairness of comparison is not affected by the weight transfer.
>
> > W2:
>
> We have added the ablation in the revised version. The comparison between SILA-0.6B and GatedDelta+Ours-0.6B (under identical training) shows the contribution of the architectural design. The comparison between models with and without the weighted loss shows the contribution of the training strategy.
> > W3:
>
> We respectfully argue that our setup provides clear evidence that the gains stem from the proposed method. First, SILA-0.6B outperforms baselines that are both larger in parameter count and trained on more data, proving gains are from our method, not scale or data. Second, GatedDelta+Ours-0.6B under the exact same training setup provides a clean ablation, confirming the gains are not from a specialized setup.
> > Q1:
>
> Here, Qwen3 serves as a strong reference for an upper-bound performance. Its robust NIAH capability stems primarily from being trained with a 32k context length. If the training length of a Transformer is not enough, it will not have robust performance on long-context tasks [[1]](https://arxiv.org/abs/2410.17980).
> > Q2:
>
> For parameter cost, SILA adds three small weights compared to Gated DeltaNet: $W_{\beta1},W_{\gamma1}\in\mathbb{R}^{d\times h}$, and $W_\alpha\in\mathbb{R}^{hd\times h}$, where d is the head dimension, h is the number of heads. So the total parameter increase is $2dh+dh^2$ per layer. For SILA-0.6B (d=64, h=16, 28 layers), the parameter increase is only 0.5M, which is negligible.
>
> For computation cost, the additional operation in SILA includes (1) using the key to retrieve the memory, and map it to gating (Eq. 5, $W_{\beta1}M_{t-1}k_t$ and  $W_{\gamma1}M_{t-1}k_t$); (2) compute the additional gate $\alpha_t$ in Eq. 6; (3) the second term of $o_t$ in Eq. 6. In our implementation, these operations lead to a 5%–10% increase in inference time.
> > Q3:
>
> It seems your question refers to Eq.4, since Eq.10 does not include a forget gate. We constructed the forgetting gate with these principles: (1) the output range is (0, 1] or [0, 1]; (2) can reach exactly 1; (3) monotonical and continuously differentiable. We didn't use Sigmoid because it cannot reach exactly 1, which may always cause rapid forgetting on long context and thus hindering performance.
> > Q4:
>
> Yes, the nonlinearity does hurt parallelism. We implementated a Triton kernel, which uses recomputation to avoid materializing all intermediate states across time steps. But in backpropagation, the kernel does need to materialize an $O(Td^2)$ space temporarily. As we shown in the ablation, the largest performance gain comes from training strategy, applying the weighted loss to Gated DeltaNet also enhances the performance, without hurting parallelism. The memory-dependent gates provide a further performance improvement at the cost of introducing nonlinearity.
> > Q5:
>
> We choose these hyperparameters empirically, without systematic sweeps. We visualize the token weights, then adjust the threshold and scaling factor to ensure that: a) tokens with long-range dependencies are assigned higher weights, and b) the resulting weight distribution remains balanced, avoiding extreme values.
> > Q6:
>
> Thank you for pointing this out. We have now included these results in the revised version.
> > Q7:
>
> To verify that SILA learns selective ignoring to improve long-context retrieval, we added a new analysis in the revised version. We evaluated models on NIAH-Word with both no-inst and strong-inst variants. SILA-0.6B shows a significantly larger performance gain than baselines after adding instructions. We visualize the activity of SILA's input gates ($\beta_t$) on an NIAH-Word sample under both no-inst and strong-inst settings (in Appendix D). In some memory heads, the input gates exhibit a spiky activation pattern. When strong instruction is added, the spike at the answer position increases notably, while other spikes are slightly suppressed. Under strong-inst, the input gate attains its highest values precisely at the query and answer positions, which is consistent with the intended behavior of selective ignoring. In contrast, such a clear activation pattern is not observed in Gated DeltaNet-1.5B. This pattern is widely observed across memory heads in SILA, indicating how selective ignoring operates in practice.

---

### Official Review · Reviewer_Roi4 · 2025-10-31

**Soundness:** 2
**Presentation:** 3
**Contribution:** 2
**Rating:** 4
**Confidence:** 4

**Summary:**

The paper re-frames long‑context retrieval in linear attention models as a memory‑writing problem and argues that strong performance often comes from specialized digit‑token shortcuts rather than general associative recall. The authors show instability in standard NIAH evaluation and recommend sample‑level haystack shuffling and also propose SILA, a linear‑attention variant that decouples recall from writing and introduces memory‑dependent gates.

**Strengths:**

1. The paper convincingly shows that linear models often “win” on NIAH by preferentially writing digits rather than learning general retrieval. This is an important finding.
2. Two concrete fixes—sample‑level shuffling and NIAH‑Word—expose the over‑reliance on digits. This yields an evaluation setup other works can adopt immediately.
3. SILA’s read‑before‑write decoupling  allows using the current token locally without committing it to memory; memory‑dependent gates use retrieved state to decide writing/forgetting.

**Weaknesses:**

1. Teacher‑dependence & compute overhead not quantified. The weighted‑loss pipeline requires per‑token attention from a reference Transformer. The paper does not report added training FLOPs/throughput or wall‑clock vs. a standard linear‑attention pretrain.
2. SILA performs an extra memory read for gating/recall each step. Please report inference speed and memory footprint vs. comparable linear baselines.
3. Unfair comparisons: SILA‑0.6B is initialized from Qwen3‑0.6B and trained on 15B FineWeb‑Edu tokens with bespoke weighting, whereas many baselines are off‑the‑shelf with different data/state sizes.

**Questions:**

N/A

---

> ### Author Response · Authors · 2025-11-23
>
> Thank you for your valuable feedback. We have revised the manuscript accordingly and address each of your points as follows:
> > W1: about teacher‑dependence & compute overhead
>
> We clarify that our method is not distillation. We do not train our model to mimic the Transformer's output distributions. Instead, we use the Transformer only once to give each token a score, creating a static dataset reused for all subsequent model training. This eliminates any ongoing computational dependency during training. Besides, there is no dependency on specific Transformer models, any Transformer with long-context capabilities will work. Quantitatively, scoring 15B tokens with Qwen3-0.6B requires ~40 hours on a single H100—a fixed, one-time cost that remains regardless of the number or scale of linear attention models to be trained. Furthermore, this cost could be reduced via inference optimizations like quantization.
> > W2: about inference speed and memory footprint
>
> We have tested the inference speed and memory footprint against the model without the extra gating/recall (corresponding to GatedDeltaNet) on H100. Over sequence lengths from 1k to 8k, the inference time of SILA is 1.05-1.10x that of the baseline, with a memory increase of <1%.
> > W3: about unfair comparisons
>
> First, to isolate the impact of initialization, we initialized a standard Gated DeltaNet model from Qwen3-0.6B and trained it with standard loss on same training data. This model exhibits poor performance on NIAH (Table 2, GatedDelta+Ours-0.6B w/o. weighted loss), demonstrating that initialization from Qwen3 does not benefit long-context capabilities. All improvements come from our architecture and weighted loss design.
>
> Second, the varied configurations of the baselines, in fact, place our model at a disadvantage. Our models have the smallest training data scale and second smallest state size among the baseline linear models, but still surpass larger models, which we consider sufficient to validate the effectiveness of our methods.

---

### Official Review · Reviewer_5kAD · 2025-11-01

**Soundness:** 3
**Presentation:** 3
**Contribution:** 3
**Rating:** 6
**Confidence:** 3

**Summary:**

This paper proposes SILA (Selective Ignoring Linear Attention), an architecture designed to enhance the long-context retrieval ability of linear attention models. SILA introduces (1) a memory-dependent gating mechanism to selectively write information into memory and (2) a weighted-loss scheme to emphasize important tokens during training. Experiments show strong gains in synthetic and benchmark long-context retrieval tasks, demonstrating better generalization to 10–100× longer contexts than training.

**Strengths:**

- Well-motivated and relevant problem: The paper tackles a key limitation of efficient attention models—poor long-range retrieval—highly relevant to long-context LLM research.

- Clear and technically novel mechanism: The selective ignoring gate and weighted-loss supervision are simple yet effective extensions that yield interpretable and consistent improvements.

- Comprehensive empirical validation: The paper provides clear component-wise ablations and visual analyses showing how the gating improves selective memory use.

**Weaknesses:**

- Limited scalability to current LLMs: SILA requires replacing the attention mechanism and retraining from scratch, making it impractical for integration into existing large pretrained models (e.g., GPT-series).

- Benchmark and scale limitations: Evaluations are restricted to mid-sized (0.6B) models and synthetic retrieval tasks, lacking validation on realistic large-scale or multi-domain benchmarks.

- Incomplete comparison: The paper does not benchmark against other recent selective-memory or efficient-attention architectures.

**Questions:**

see weeknesses.

---

> ### Author Response · Authors · 2025-11-23
>
> Thank you for your valuable feedback. We have revised the manuscript accordingly and address each of your points as follows:
> > W1: Limited scalability to current LLMs
>
> Linear attention architectures (e.g., Mamba, DeltaNet, RWKV, and our SILA) represent a new paradigm different  from the Transformer, which inherently requires training from scratch. We would like to clarify that this is standard for architectural innovations, not a limitation.
> > W2: Benchmark and scale limitations
>
> We agree that scaling to larger models is an important future direction. While our current computational resources preclude industrial-scale validation, our evaluations at the 0.6B scale are sufficient for our core claim: selective ignoring enhances long-context retrieval. Crucially, our 0.6B model outperforms larger models which have more parameters and training tokens (Table 2), demonstrating the effectiveness of our method. We also have experiments on commonsense reasoning tasks in Table 3, showing that the improvements for long-context retrieval do not sacrifice the model's general capabilities.
> > W3: Incomplete comparison
>
> We have given a detailed comparison and analysis of recent selective memorization and extrapolation methods for linear attention models in Appendix B. Furthermore, the baseline architectures adopted in this paper (Mamba2, Gated DeltaNet, RWKV7) are the most mainstream and advanced ones which have been widely recognized by both academia and industry. Additionally, our model also outperforms the latest Mamba3 (under review at ICLR 2026) on long-context retrieval.

---

### Official Review · Reviewer_iNnf · 2025-11-03

**Soundness:** 3
**Presentation:** 3
**Contribution:** 2
**Rating:** 4
**Confidence:** 2

**Summary:**

Motivated by the limitations of existing linear attention methods that fail to perform robust long-context retrieval due to the fixed memory capacity, the paper proposes a novel linear attention method that learns to selectively ignore irrelevant tokens for long-context retrieval tasks and attend to important instruction tokens. The proposed method, Selective Ignoring Linear Attention (SILA), shows improvement on needle-in-the-haystack tasks with better memory utilization efficiency compared with prior methods.

**Strengths:**

1. The paper is motivated by the limitations of existing linear attention methods on robust long-context retrieval tasks, with well-designed controlled experiments, and the proposed method is designed to solve the identified issues of prior methods.

2. The paper conducts extensive experiments and compares the proposed methods over several strong baselines from the literature. Given similar training token sizes and model sizes, models trained with the proposed method outperform strong baselines, such as RWKV7 and Gated DeltaNet, by a large margin, especially on NIAH-word, demonstrating better robustness.

3. The paper further demonstrates the efficiency of the proposed method compared with other linear attention methods and also shows the general reasoning capabilities of the method, adding empirical strengths to the method.

**Weaknesses:**

1. The evaluation of long-context retrieval is limited to NIAH and its variants. The proposed method is only compared with baselines on NIAH-1, NIAH-2, and NIAH-Word in Table 2. On the passkey retrieval task, SILA is not compared with any other baseline; on the in-context recall task from MAD-Lab benchmark, SILA is only compared with baselines in the setting using 2-layer shallow models instead of the 0.6 B model. The empirical strength of SILA on long-context retrieval needs to be further validated by comparisons with other methods on more benchmarks, such as MAD, Multi-query associative calls (MQAR), and RegBench.

2. There lacks analysis of different components of SILA. There is no ablation of the loss design or gate design of SILA, and it is unclear which part of the proposed method contributes most to the empirical improvements on NIAH.

3. Particularly, there is no analysis of how SILA addresses the memorization issues of other linear attention methods on NIAH: Is there evidence for SILA models not biasing towards digit tokens and performing general retrieval tasks, beyond the results on NIAH-Word? Is there evidence for SILA models correctly attending to the instruction tokens for the strong instruction variants of NIAH?


That being said, I'm willing to adjust the scores if the authors can provide more analysis results, especially the ones mentioned above.

**Questions:**

1. In Table 2, are the NIAH-1 and NIAH-2 datasets with or without sample-level shuffling, following findings from Figure 1(b)?

---

> ### Author Response · Authors · 2025-11-23
>
> Thank you for your valuable feedback. We have revised the manuscript accordingly and address each of your points as follows:
> > W1: aboud evaluation
> 1) We would like to clarify that the NIAH-1 task is the passkey retrieval task with only minor prompt variations. Therefore, the NIAH-1 results in Table 2 directly address your request for a passkey retrieval comparison.
> 2) Regarding the MAD-Lab benchmark, it is specifically designed as a model prototyping and diagnostic tool, not for evaluating large-scale models. Our comparison using shallow models strictly follows the intended usage of the MAD-Lab benchmark. Conducting this evaluation on 0.6B models is not standard practice in the community.
> 3) The key strength of SILA is the selective ignoring capability, which is triggered by instructional cues. However, benchmarks like MQAR present the query after the entire context without any prior instruction. This setup necessitates memorizing the entire context and inherently bypasses the "selective ignoring" mechanism, making these benchmarks less relevant for evaluating SILA's primary contribution. Nevertheless, to address the reviewer's concern, we have conducted experiments on MQAR. As SILA is adapted from Gated DeltaNet, it performs at least on par with the baselines. As shown in the following table, it shows slightly better performance, confirming that our method maintains robustness on tasks outside its primary design scope.
>
> MQAR results with seqlen=512, kv_pairs=64:
> |  | dim=32 | dim=64 | dim=128 |
> | :--- | :--- | :--- | :--- |
> | DeltaNet | 0.313 | >0.99 | >0.99 |
> | Gated DeltaNet | 0.411 | 0.961 | >0.99 |
> | SILA | 0.441 | >0.99 | >0.99 |
> > W2: about ablation
>
> We have added the ablation in the revised version. The comparison between SILA-0.6B and GatedDelta+Ours-0.6B (under identical training) shows the contribution of the architectural design. The comparison between models with and without the weighted loss shows the contribution of the training strategy. Results show that the largest improvement stems from the training strategy, while the architectural design, although less impactful, also yields a certain degree of enhancement.
> > W3: about memorization
>
> For the first question, We clarify that SILA does not fundamentally solve the inherent bias towards digit tokens, which is largely rooted in the statistical biases of the pre-training data. The primary goal of our NIAH-Word experiment is not to claim the solution of this bias, but to demonstrate that the classic digit-based NIAH benchmark can not fully reflect a model's general retrieval capability.
>
> For the second question, we do have evidence that SILA attends to instruction better. In the revised version (also listed below), we tested SILA and baseline models on NIAH-Word with both the no-inst and strong-inst variants, and compared the performance gain from instructions for each model. SILA has a significantly larger performance enhancement than baseline models after adding instructions, which is a supporting evidence for its better utilization of instructions. We also provide a visualization of memory writing patterns in Appendix D, which explains how the model uses selective memory writing to enhance retrieval performance.
> | Model              | no inst 2k | no inst 4k | strong inst 2k        | strong inst 4k        |
> | :--- | :--- | :--- | :--- | ---- |
> | Mamba2-780M        | 28.4       | 0.0        | 37.4 ($\uparrow$9.0)  | 0.0 ($\uparrow$0.0)   |
> | DeltaNet-1.3B      | 15.6       | 8.0        | 20.4 ($\uparrow$4.8)  | 6.8 ($\uparrow$0.0)   |
> | GatedDeltaNet-1.5B | 29.6       | 9.8        | 37.8 ($\uparrow$8.2)  | 11.6 ($\uparrow$1.8)  |
> | RWKV7-0.4B         | 18.0       | 9.6        | 26.2 ($\uparrow$8.2)  | 10.2 ($\uparrow$0.6)  |
> | SILA-0.6B          | 31.0       | 10.6       | 63.6 (**$\uparrow$32.6**) | 25.8 (**$\uparrow$15.2**) |
> > Q1: about NIAH settings
>
> In Table 2, all datasets are with sample-level shuffling, so that the benchmark results can be more stable and reliable.

---

### Meta-Review · Area_Chair_Yt6u · 2026-01-07

**Summary:**

This paper proposes SILA (Selective Ignoring Linear Attention) to address limited long-context retrieval in linear attention models through memory-dependent gating and weighted loss training. The authors identify digit token bias in existing models via diagnostic experiments (NIAH-Word benchmark) and demonstrate 20× context length extrapolation. Reviewers acknowledge the well-motivated problem and technical contributions but raise concerns about limited scale (0.6B only), constrained experimental settings (15B training tokens vs 300-2000B for baselines), and incomplete benchmark coverage.

**Reviewer Concerns:**

While the rebuttal addressed some concerns, comparisons still feel unfair: SILA uses Qwen3 initialization, while many baselines are off-the-shelf, so the superiority claims remain partially confounded. The authors tried to justify the experimental design choice by adding one experimental result, which is not sufficient.

**Reviewer Scores:**

Mixed to negative: All reviewers scored 4 except one who scored 6. Reviewers acknowledge the paper's well-motivated problem and interesting approach but raise significant concerns about experimental completeness, fair comparisons, and scalability. The core issues center around: (1) limited benchmark diversity beyond NIAH variants, (2) missing ablation studies, (3) unfair baseline comparisons due to different training setups, and (4) constrained experimental settings that limit generalizability. The rebuttal couldn't resolve most of the major concerns, so not much score increase is expected.

---

### Decision · Program_Chairs · 2026-01-26

Reject